# A Simulation Study on the Influence of Street Tree Configuration on Fine Particulate Matter (PM$_{2.5}$) Concentration in Street Canyons

Junyou Liu and Bohong Zheng *

School of Architecture and Art, Central South University, Changsha 410083, China
* Correspondence: zhengbohong@csu.edu.cn

**Abstract:** Because motor vehicles emit a large amount of PM$_{2.5}$ pollution, traffic-related emissions have always been an important part of PM$_{2.5}$ pollution. To better understand the influence of street trees on traffic-related PM$_{2.5}$ pollution, our study focused on camphor trees, common evergreen urban street trees in central and southern China. We used ENVI-met for the simulation of PM$_{2.5}$ pollution and to build a model to show the distribution of PM$_{2.5}$ pollution along a section of Xinyao North Road in downtown Changsha City in central China. Based on this model, we constructed four other models with different heights, quantities, and distances between street trees, where each model had high feasibility and aimed to determine how these affect the PM$_{2.5}$ concentration on the designated block. We performed simulations within different time frames in the year. We found that the wind can promote the diffusion of PM$_{2.5}$ in the street canyon. Too dense a distribution of tall street trees will have a negative impact on PM$_{2.5}$ concentration in street canyons. A moderate distance between street trees is conducive to the dispersion of pollutants. Because the crown of 5 m high street trees is small, its negative impact on the dispersion of wind and PM$_{2.5}$ is relatively small, so further increasing the number of 5 m high street trees in street canyons with densely distributed tall street trees will have only a little more negative impact on PM$_{2.5}$ concentration in street canyons. The PM$_{2.5}$ concentration in the street canyon is generally better when the street trees are 5 m long, even if the number of 5 m high street trees is relatively large. Although the crown size of 15 m high street trees is larger than that of 10 m street trees, the vertical distance between the canopy of 15 m high street trees and the ground is usually greater than that of 10 m high street trees. The distance between the canopy of 15 m high street trees and the breathing zone is usually greater than that of the 10 m street trees. Longer distances lead to a weakening of its impact on PM$_{2.5}$. When the 15 m high and 10 m high street trees are more scattered in the street, their effects on the PM$_{2.5}$ concentration at the height of the breathing zone (1.5 m) are generally similar.

**Keywords:** street canyons; street tree configuration; PM$_{2.5}$; simulation

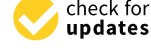



## 1. Introduction

Fine particulate matter, also known as PM$_{2.5}$, is composed of particles in the atmosphere with a diameter of less than or equal to 2.5 μm and is one of the most prevalent air pollutants in many countries. The concentration of PM$_{2.5}$ is positively correlated with the prevalence rate of cardiovascular and respiratory diseases among residents [1,2]. According to estimates by the World Health Organization (WHO) from 2019, PM$_{2.5}$ leads to the premature death of approximately 4.2 million people per year [3]. Some researchers estimate that when the concentration of PM$_{2.5}$ increases by 10 μg/m$^3$, the cardiorespiratory death rate increases by 6% [4]. However, even when only a small concentration of PM$_{2.5}$ is in the air, it can still negatively affect the human body [5].

Traffic is one of the primary sources of PM$_{2.5}$ pollution and one of the three main factors affecting the concentration of PM$_{2.5}$ pollution on urban roads, along with vegetation

characteristics and meteorological factors. In densely built-up areas, there is very limited air exchange between the ground level and the above-roof level, and it is difficult to dilute and remove traffic pollutants near the ground, resulting in the accumulation of a relatively high concentration of pollutants on the road. Street greening can have a variety of positive effects, such as improving air quality, mitigating the heat island effect, protecting biodiversity, reducing noise pollution, and preventing stormwater overflow. The construction of street greening has been considered one of the main countermeasures to reduce $PM_{2.5}$ pollution from the traffic source due to its aerodynamics, deposition, and resuspension effects on the fine particulate matter [2,6–8]. Some researchers have conducted research on the influence of road green space on particulate matter. This article attempts to further explore the number, height, spacing, and canopy characteristics (crown height and crown width, etc.) of street trees on $PM_{2.5}$ concentrations in street canyons based on existing research [2,8–10].

The concentration of $PM_{2.5}$ can be affected by meteorological factors such as temperature, humidity, and wind speed [2,11]. Strong winds can cause the resuspension of many large-particle-size particulate matters while accelerating the diffusion of particulate matters and diluting small-particle-size particulate matters. A lower ambient temperature and a higher relative humidity aid in the nucleation of particles and increase the number and concentration of particles [11,12].

Different street green space patterns can significantly affect the concentration of $PM_{2.5}$ [13]. Hong et al. (2017) [14] investigated the impact of street trees on the dispersion of $PM_{2.5}$ in terms of crown shape, leaf area density (LAD), and street canyon aspect ratio (H/W). The LADs included three scenarios of 0.5, 1.5, and 2.5 $m^2/m^3$, and the aspect ratios included three scenarios of 0.5, 1.0, and 2.0. A total of 27 scenarios were constructed in the study, and it was determined that the scenarios where the H/W = 1.0 and LAD = 1.5 were the most effective at capturing $PM_{2.5}$. Some researchers found that a higher LAD was associated with a higher deposition rate under natural conditions [15]. However, when the LAD is significantly greater than in a natural setting, the deposition rate of particulate matter decreases as the LAD increases [16]. In terms of crown shape, the conical canopy has the highest rate of $PM_{2.5}$ reduction, followed by the spherical canopy, and the cylindrical canopy has the lowest [14]. Depending on the properties of the planting site, the effects of plant height on air quality vary [17]. Some researchers suggested that the moderate distance between street trees may contribute to the diffusion of $PM_{2.5}$ in the atmosphere, allowing the green space to more effectively reduce $PM_{2.5}$ [1]. However, Gromke et al. (2016) [18] and Li et al. (2016) [19] noted that a row of continuous hedgerows without openings and gaps could improve air quality in a street canyon. Liang et al. (2016) [20] discovered that the size of leaf pores and the groove proportion were positively correlated with a plant's ability to capture $PM_{2.5}$. Due to their rich leaf morphology, broad-leaved trees have a greater ability to capture $PM_{2.5}$ with a single leaf, whereas needle-leaved trees have a greater total leaf area, so they have a greater ability to capture $PM_{2.5}$ as a single tree. However, Chen et al. (2021) [7] discovered in their study that different barrier parameters of street canyon plants (species and height) had no significant impact on the concentration of $PM_{2.5}$. The density of branches also influenced the deposition and accumulation of particulate matters [15]. In general, optimizing the street green space pattern can be regarded as an effective way to reduce the $PM_{2.5}$ concentration [15,21,22].

Urban street green space can reduce $PM_{2.5}$ pollution in the atmosphere through four regulatory and control mechanisms: deposition, retardation, adhesion, and absorption. In short, deposition is the changes in the aerodynamic state of green spaces that causes $PM_{2.5}$ to fall on plants through aerodynamic effects such as gravity deposition. Retardation is the changes in the movement trajectory of the airflow in the green space caused by the passage of $PM_{2.5}$, which alters the movement energy and mode and causes particulate matters to remain in the green space. Meanwhile, adhesion is the capture of $PM_{2.5}$ via the villi, folds, and special secretions on the surfaces of leaves, branches, stems, and other parts of plants, and absorption is the process by which $PM_{2.5}$ is absorbed into the plant

tissue through physiological effects and is degraded during the metabolic process, thereby removing $PM_{2.5}$ from the atmosphere [23].

Street greening can also increase the concentration of $PM_{2.5}$ in a street canyon; street trees can impede the exchange of ambient air and the diffusion of automobile exhaust gases, thereby increasing the concentration of air pollution [11,13]. $PM_{2.5}$ can float in the atmosphere for an extended period and may not settle easily, making its deposition process almost negligible [2]. Several field research studies have confirmed that street green spaces increase the concentration of $PM_{2.5}$ in the atmosphere [2,24] and that street green spaces can have both positive and negative effects on the surrounding $PM_{2.5}$ pollution, depending on their design characteristics [15,25].

This study aims to explore the effects of the distance, number, height, and canopy characteristics (crown height and crown width, etc.) of trees on the concentration of $PM_{2.5}$, focusing on the following issues: (1) Is a continuous or intermittent arrangement of street trees more effective at reducing the concentration of $PM_{2.5}$? (2) What is the optimal distance between street trees where the positive effect on $PM_{2.5}$ pollution is the greatest and the negative effect is the smallest? (3) Is there a specific rule regarding the distance between street trees and its effect on the concentration of $PM_{2.5}$?

## 2. Materials and Methods

### 2.1. Overview of Study Area

Changsha, located in south central China between 27°51′ N and 28°40′ N and between 111°53′ E and 114°15′ E, is the capital of Hunan Province and has a subtropical monsoon climate. Figure 1 shows the geographical location of Changsha. The city experiences extreme temperature fluctuations in the spring, receives a lot of rain in early summer, maintains long periods of high temperatures in the autumn, and is very cold in the winter. In 2021, the city had a permanent population of 10.24 million [26]. Statistical data from 2018 revealed that resuspended dust, mobile sources, and industrial processes accounted for 26.7%, 20.3%, and 16.1% of the total $PM_{2.5}$, respectively. From 2018 to 2021, the annual average concentration of $PM_{2.5}$ in Changsha was 44.08, 47.17, 41.25, and 42.25 μg/m$^3$ [27–29], remaining above the Level 2 limit of the concentration of $PM_{2.5}$ in China of 35 μg/m$^3$ [30,31] and the WHO's reference value of 5 μg/m$^3$ [32]. A road section at the intersection of Xinyao Road and Furong South Road in downtown Changsha was selected as the main area for research.

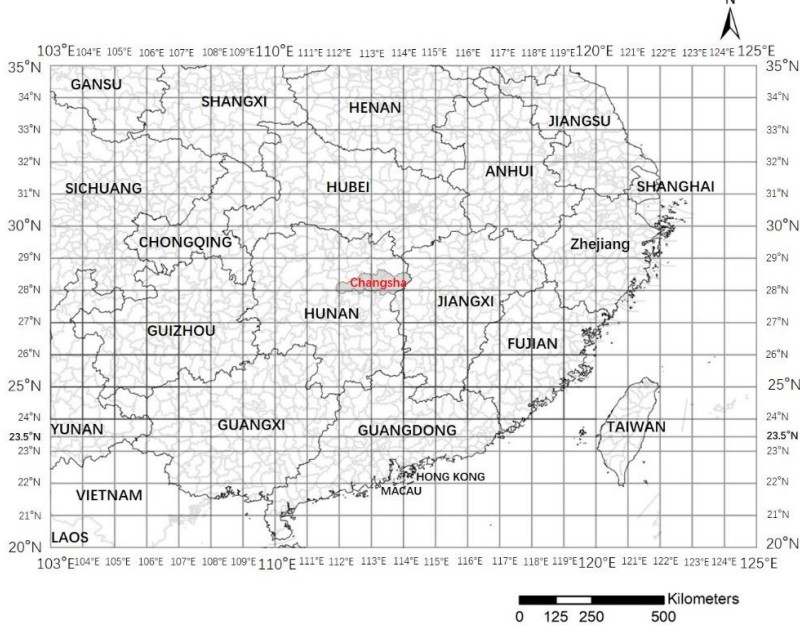

**Figure 1.** A geographical location map of Changsha City.

*2.2. Overview of ENVI-Met*

The ENVI-met was designed in 1994 by Professor Michael Bruse from Germany [33]. Since then, his team has been dedicated to developing and expanding the scientificity of the software, which has been confirmed by the scientific results obtained from more than 3000 independent studies and technical publications [34]. ENVI-met can be used to calculate the leaf temperature in model grids (horizontal resolution is typically between 1 and 10 m) while taking into account the photosynthesis rate, water use efficiency, and local microclimate conditions to analyze in a complicated environment using computational fluid dynamics [35]. The applications of air pollution simulation have gradually increased. It can simulate the release, diffusion, and deposition of pollutants, one of which is $PM_{2.5}$.

ENVI-met adopts the standard convection–diffusion equation [36] for the diffusion simulation of gases and particles:

$$\frac{\partial \chi}{\partial t} + u\frac{\partial \chi}{\partial x} + v\frac{\partial \chi}{\partial y} + w\frac{\partial \chi}{\partial z} = \frac{\partial}{\partial x}\left(K_x\frac{\partial \chi}{\partial x}\right) + \frac{\partial}{\partial y}\left(K_x\frac{\partial \chi}{\partial y}\right) + \frac{\partial}{\partial z}\left(K_x\frac{\partial \chi}{\partial z}\right) + Q_x(x,y,z) + S_x(x,y,z) \quad (1)$$

where $\chi$ refers to the component of a gas or particle in the atmosphere as simulated in the experiment, measured in the unit of $\left[mg(\chi)kg^{-1}(air)\right]$; $Q_x$, measured in $\left[mgkg^{-1}s^{-1}\right]$, and $S_x$ refer to the pollution source and deposition type (deposition or chemical reaction), respectively.

The $Q_x$ pollution sources can be divided into point sources, line sources, area sources, and volume sources. The traffic pollution source falls under the line sources. The $S_x$ deposition types include particles deposited from the top layer, particles subject to gravity deposition, and the total loss of particles caused by the deposition on the surface of the leaf. The effect of gravity on pollutants, the effect of plants on pollutants, and the microclimate (temperature, humidity, wind direction, wind speed, etc.) are all considered in ENVI-met [36,37].

*2.3. Field Survey*

We made an investigation with eight roads in the central area of Changsha. The main data collected included tree height, canopy diameter, LAI, distance between street trees, and others. A roller diastimeter was used to measure the distance between street trees. The precision is the absolute value of the difference between the measured value, and the actual value is normally no more than 0.5 percent of the measured value. Although there are some shrubs in the section of road selected, they were not used as a variable in the study. The names of the plants were judged using the Xingse software (version 3.14.12) and later confirmed by searching for further information and asking experts. The arbor height of related trees was measured using the direct-reading height indicator CGQ-1 (Harbin Optical Instrument Factory Ltd., Harbin, China). Because the shrubs and grasses were short, the height of a given set of shrubs and grasses was measured with a tape measure. We used the Heimiview plant canopy analysis system (Delta-T Devices Ltd., Cambridge, UK) to measure the LAI of arbors. The system has been used often and is believed by many researchers to be reliable for assessing the LAI analysis results of plants [38–40]. For the determination of the LAI of relatively short shrubs and grasses, we measured the leaf area of some plants with the LD-YMJ-B (Shandong Laiende Intelligent Technology Co., Ltd, Shandong, China), calculated the floor area, and finally determined the LAI. The error of the LD-YMJ-B in measuring the leaf area of plants was ±2%.

The height of tall trees in these eight road sections was about 15 m, and that of short trees was about 5 m. The average height of street trees in many road sections was about 10 m. We built models of street trees with 3 different heights, including 5 m, 10 m, and 15 m, for study. The average height of street trees in the road section of Xinyao Road investigated by us was about 10 m. According to the actual conditions of this road section, we built Model 1 to explore the impact of street trees on $PM_{2.5}$ in street canyons when the number of street trees in the study area was 42, the height of trees was 10 m, the canopy diameter of the trees was 7 m, and the distance between most street trees was 6–9 m. In order to control

the impact of factors such as the characteristics of surrounding buildings, road width, traffic flow, and meteorological conditions on our research, we built other models based on the change of the street tree characteristics of model 1. We found that the distance between trees could be up to 9–12 m in some road sections (especially some wide road sections). On this basis, we built Model 2 with a small number of street trees (26) and a big distance. The distance between street trees in this model was often 9–12 m. The height of trees and canopy diameter in Model 2 are the same as those in Model 1. There are many tall and short street trees simultaneously in some road sections, and street trees are closely distributed. We built Model 3 in which 10 m and 5 m street trees were alternatively distributed, there were a large number of street trees (71), and they were closely distributed. The distance between street trees was often 3 m. In the process of investigation, we found that the height of some street trees was only 5 m. On this basis, we further thought about the concentration of $PM_{2.5}$ in street canyons if the heights of all street trees were 5 m. Because the canopy diameter of the 5 m street trees was small, when the model was built, we set the number of 5 m street trees as 55, and the distances between the street trees were mostly around 6 m. Some of the road section was characterized by a big distance between tall trees. On this basis, we built Model 5 with a tree height of 15 m, the number of street trees of 26, and a distance between most street trees of 9–12 m.

We also conducted more detailed field surveys on the Xinyao North Road. We collected data at periods from 14:59 to 16:00, 16:59 to 18:00, and 20:59 to 22:00 on 24 August 2022; from 8:59 to 10:00, 9:59 to 11:00, 14:59 to 16:00, and 16:59 to 18:00 on 28 September 2022; from 14:59 to 16:00 and 20:59 to 22:00 on 22 December 2022; and from 14:59 to 16:00 on 24 December 2022. This allowed us to research the area based on the conditions at different periods throughout the year. We first measured the $PM_{2.5}$ concentration in the first minute of each of the above time periods and took it as the background concentration. We then measured the $PM_{2.5}$ concentration in the last minute of the above time periods and used them to calculate the error between the measured values and the simulated values. We used DustTrack 8530 aerosol detector (TSI, Saint Paul, MN, USA) to measure the $PM_{2.5}$ concentration. Under the premise of proper calibration, the linear regression $R^2$ value between the $PM_{2.5}$ value measured by TSI DustTrack 8530 and stationary reference instruments (standard value) can achieve above 0.9, and the RMSE value can be within 3.5 μg/m$^3$ [41]. In addition, hourly traffic flow statistics were counted using mechanical counters, and the emissions of $PM_{2.5}$ for each hour were calculated according to the related computational formula in ENVI-met. The hourly traffic flow is determined by taking the actual measurement of the traffic flow for 10 min within that hour and multiplying it by 6.

The hourly average wind direction, wind speed, temperature, and humidity data were measured using the Kestrel 5500 (Nielsen-Kellerman, Upper Chichester, PA, USA). The accuracy of Kestrel 5500 measuring temperature, humidity, wind direction, and wind speed is 0.5 °C, 2% RH, 5°, larger than 3% of reading, least significant digit or 20 ft/min speed, respectively [42]. Moreover, the weather meter has been applied in experiments with high measurement accuracy requirements. For example, it is used by Ciptaningayu et al. (2020) [43] as a measurement tool for microclimate environment in the field research on urban particulate matter and by Yuan et al. (2021) as an instrument to measure microclimate conditions in a scientific study on the existence of Brown Carbonaceous Tarball in the Himalayan Atmosphere [44].

*2.4. Simulation Study and Error Analysis*

ENVI-met was used to simulate the distribution of $PM_{2.5}$ in the studied road section. The associated data input was collected during the measurement, and the background concentration of $PM_{2.5}$ pollution input during the simulation was the average concentration of $PM_{2.5}$ of the previous minute. Model 1 is a model built according to the actual conditions of the road section. Then, based on the conditions of the surveys of the urban roads, four models were built for comparative studies. Before each simulation, the corresponding

meteorological parameters and the background concentration of $PM_{2.5}$ were input, and the distribution of $PM_{2.5}$ pollution in the study area over the next hour was simulated.

In order to determine the emission factors of various motor vehicles, we referred to the relevant national standards, the standards of Changsha, the relevant research literature, and the actual conditions of this road section. In addition, based on relevant data, we applied COPERT V to determine the emission factors of various motor vehicles.

A fixed point was selected in the study area, and the hourly concentrations of $PM_{2.5}$ were measured. After completing the simulation, the simulated concentrations of $PM_{2.5}$ at the corresponding point were found. The error analysis was conducted based on the measured and simulated $PM_{2.5}$ concentration.

## 3. Result and Analysis

### 3.1. Data Measurement

We selected some road sections of eight different roads and measured the distance between street trees. The schematic diagram of the road section and the distance between street trees are shown in Figure 2 below. Table 1 below further summarizes the distance between street trees in each road section. The average distances between the street trees in the eight selected road sections are mostly between 8 and 12 m. The street trees alongside the urban main roads in the surveyed road sections are generally tall (many 10–15 m trees) and have a large distance between trees (around 8–10 m distance), while the street trees along the secondary roads are relatively short (many 5–10 m trees) and have a smaller distance between adjacent street trees on the same side of the road. The large distances between street trees in the table appear primarily at an intersection or a T-junction. The small distance between street trees appears primarily when between two adjacent trees (one is short, and one is tall), making it possible to make full use of the space around street trees.

**Table 1.** Characteristics of the distance between street trees on the road section studied.

| Road Section | Average Distance between Adjacent Street Trees (m) | Maximum Distance between Adjacent Street Trees (m) | Minimum Distance between Adjacent Street Trees (m) |
|---|---|---|---|
| Road section of Xinyao Road | 9.2 | 64.8 | 1.7 |
| Road section of Furong South Road | 11.9 | 54.8 | 7 |
| Road section of Shaoshan South Road | 9.3 | 21.5 | 5.8 |
| Road section of Zhutang West Road | 4.2 | 7.5 | 2.8 |
| Road section of Xiangzhang Road | 10.7 | 43.1 | 6.3 |
| Road section of Yatangchong Road | 7.9 | 26.6 | 4.3 |
| Road section of Mulian West Road | 9.6 | 23.7 | 6.8 |
| Road section of Linda Road | 5.9 | 7.8 | 4.3 |

The concentration of $PM_{2.5}$ and the related microclimate data were measured in or near the study area at different periods of time. The instrument was placed about 0.3 m above ground. This height is close to the height of the exhaust pipe of motor vehicles, which is also the height from which the $PM_{2.5}$ pollution is emitted from motor vehicles. The hourly average temperature, humidity, wind direction, and wind speed data were measured via the Kestrel 5500 anemometer.

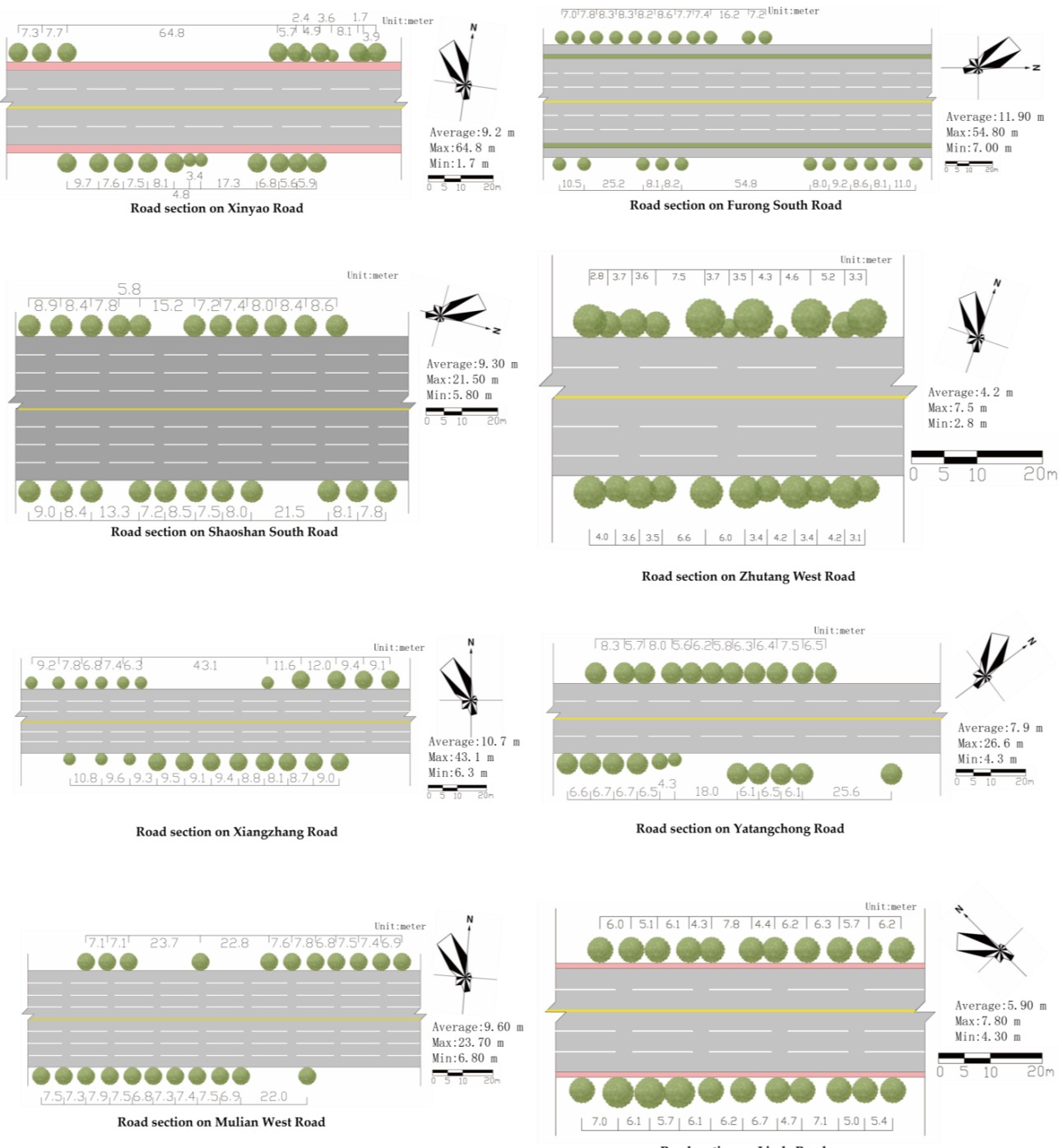

**Figure 2.** Schematic diagram of the distribution of street trees in each road section.

The traffic flow for each hour counted and estimated by us through the mechanical manual counter is shown in Table 2. The traffic volume of each time period on August 24 was estimated based on the relevant information on the traffic volume of the main road in Changsha City and the traffic volume of other comparable roads in the same time period. The traffic volume of each time period on September 28, December 22, and December 24 was estimated based on the measured traffic volume over 10 min for every hour.

**Table 2.** Statistics and estimation of traffic flow in each time period.

| Time | Estimated Value of Total Traffic Flow within 1 h |
|---|---|
| From 15:00 to 16:00 on 24 August 2022 | 2109 |
| From 17:00 to 18:00 on 24 August 2022 | 1996 |
| From 21:00 to 22:00 on 24 August 2022 | 729 |
| From 9:00 to 10:00 on 28 September 2022 | 1974 |
| From 11:00 to 12:00 on 28 September 2022 | 1704 |
| From 15:00 to 16:00 on 28 September 2022 | 1512 |
| From 17:00 to 18:00 on 28 September 2022 | 2676 |
| From 16:00 to 17:00 on 22 December 2022 | 1278 |
| From 21:00 to 22:00 on 22 December 2022 | 870 |
| From 16:00 to 17:00 on 24 December 2022 | 1512 |

The distances between adjacent street trees on either side of the street were measured (including adjacent street trees on both sides of the T-junction) using a mechanical roller diastimeter, and the corresponding measurements were used as a reference in the building of Model 1, which ensured that the distance between trees in Model 1 closely matched the actual conditions. The distance between two adjacent street trees on the same side of the road is usually between 7 and 8 m. For the section of Xinyao North Road, the average distance was calculated to be 9.2 m. The street trees were determined to be camphor trees. The height of a given set of representative trees of different species was measured using the direct-reading height indicator CGQ-1. The average height of the measured trees will be used to represent the height of this kind of tree. While the height of some shrubs was measured with a tape measure, and each of their average heights was calculated to represent the height of these shrubs. The crown width of a given set of representative trees of different species was measured using a roller diastimeter, based on which the average crown width of various trees in the study area was estimated. Table 3 shows the characteristics of related plants, which were used to build different plant models as part of the simulation model. The LAI of each plant species was determined using the Hemiview plant canopy analysis system. A certain number of plants were selected to measure their LAI, and the average of the obtained LAI of different plants of the same species of plants was used to represent the LAI of such plants. Since *Ligustrum lucidum* and *Hypericum monogynum* L. are short, a given number of plants were selected in this study. The LAI is calculated by dividing the one-sided leaf area ($m^2$) by the ground surface area ($m^2$) [45,46]. The LAIs of evergreen trees are set as consistent throughout the year. We determined the LAD at each height of the plant canopy according to the empirical relationship between LADs at different heights, as indicated in a related study [47]. The LAD was used to build tree models. There were no 5 m or 15 m camphor trees in the section of Xinyao North Road; the 5 m or 15 m camphor trees were built according to the 5 m and 15 m camphor tree models from the streets neighboring the study area.

**Table 3.** Related characteristics of various plants in the simulation model determined from measured data.

| Vegetation | Plant Height (m) | Crown Width (m) | LAI ($m^2/m^2$) |
|---|---|---|---|
| *Cinnamomum camphora* | 18 | 12 | 3.2 |
| *Cinnamomum camphora* | 15 | 9 | 3.0 |
| *Cinnamomum camphora* | 10 | 7 | 3.3 |
| *Cinnamomum camphora* | 5 | 3 | 3.6 |
| *Osmanthus fragrans* | 10 | 7 | 3.3 |
| *Ligustrum lucidum* | 0.55 | 0.39 | 2.4 |
| *Hypericum monogynum* L. | 0.55 | 0.39 | 2.3 |

The recent research of some Chinese scholars on motor vehicle emission factors and the motor vehicle pollution emission standards in China [48–52] were referred to in order to determine various vehicle emission factors. Furthermore, the emission factors of various motor vehicles were simulated in COPERT V based on the collected data, such as average speed, meteorological data, and types and quantities of various motor vehicles. The emission factors for passenger cars, light-duty vehicles, heavy-duty vehicles, and buses are 0.017, 0.015, 0.062, and 0.073 g/km per vehicle, respectively. The $PM_{2.5}$ pollution source model was built in the DB manager of ENVI-met.

### 3.2. Building the Models

This study area is located at the Xinyao North Road within the central urban area of Changsha [49]. First, aerial photographs of UAVs were imported into the AutoCAD software (version 2008), and the planar model of the study area was drawn. Then, a 3D model (Model 1) was built based on the actual condition of the planar map in SketchUp (version 2020). The tree heights in the model are all set to 10 m. The spacing between adjacent street trees on the same side of the street is mostly within 6–9 m. The number of street trees within the red-dashed areas in the schematic diagram, as shown in Model 1 in Figure 3, is 42.

Based on our field survey, we built four other models which have common street tree heights and distances. They are Models 2–5. In Model 2, the tree height within the red-dashed area is 10 m. The spacing between adjacent street trees on the same side of the street is mostly 9–12 m. The number of street trees within the area in the model is 26. There are both 10 m tall street trees and 5 m tall street trees in Model 3. The spacing between street trees is mostly 3 m. The heights of the street trees in the corresponding areas in Model 4 are mainly 5 m. The number of street trees within the red-dashed areas in the model is 42. The distances of street trees are mainly 6 m. In Model 5, there are 26 street trees which all have a height of 15 m. The distance between adjacent street trees on the same side of the street is mostly 9–12 m. The schematic diagrams of the five models are shown in Figure 3. Table 4 below shows more detailed information about the five models.

**Table 4.** Introduction to 5 different configuration models of street trees.

| Model Number | Street Tree Height (m) | Distance between Two Neighboring Trees on the Same Side of the Road (m) | Number of Street Trees within the Red-Dashed Areas | LAI |
|---|---|---|---|---|
| Model 1 | 10 | Mostly 6–9 | 42 | 3.3 |
| Model 2 | 10 | Mostly 9–12 | 26 | 3.3 |
| Model 3 | 10 and 5 | Mostly 3 | 71 | The LAI for all 10 m tall trees is 3.3, and 3.6 for all 5 m tall trees |
| Model 4 | 5 | Mostly 6 | 55 | 3.6 |
| Model 5 | 15 | Mostly 9–12 | 26 | 3.0 |

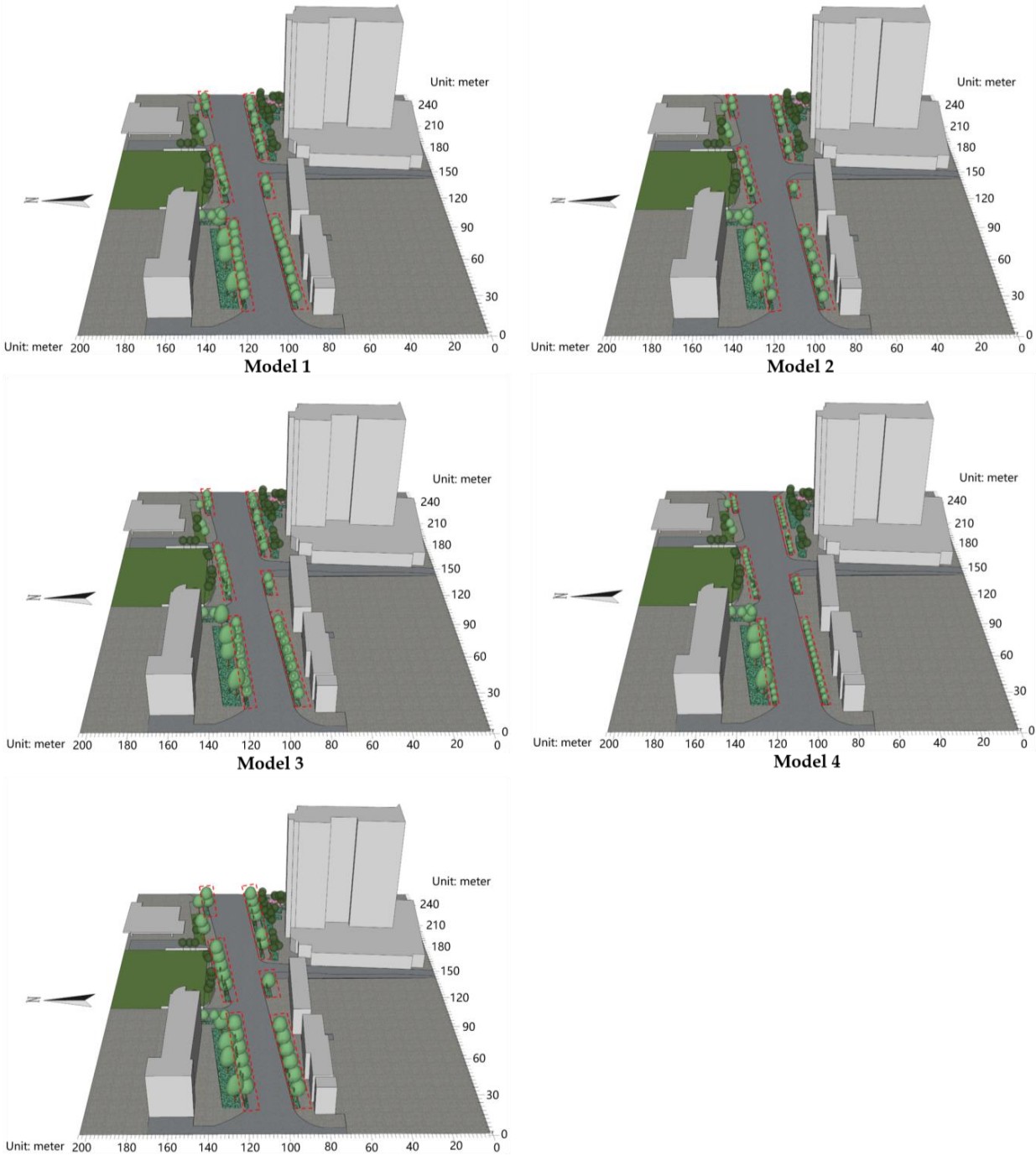

**Figure 3.** Schematic diagram of 5 different configuration models of street trees.

The models established In SketchUp were converted into models that could be simulated in ENVI-met through the INX function in SketchUp, which was also used to determine the ground, soil, and building materials, among others. Different 3D tree models were built using the Albero module in the ENVI-met. The differences between the street trees in the five models are presented in the red-dashed areas in Figure 3. The plant model could only be added in the ENVI-met but not in SketchUp. The related plant model was added according to the position information of the plant model in SketchUp.

The general condition of the models is shown in Figure 3. The five models had the same overall volume. The number of grids occupied by the models on the X-, Y-, and Z-axes was 137 ×152 × 25, where the length of a single grid on the X-axis was 3 m, the

length of the unit grid on the Y-axis was 2 m, and the length of a single grid on the Z-axis was 3 m. Therefore, the total volume of each model was $411 \times 304 \times 75$ m$^3$. In the vertical direction, telescoping was started after a height of 20 m was reached at a 10% telescoping factor. Because there was adequate space between the top and boundary of the study area and the building, no nested grid was added.

*3.3. Error Analysis*

We measured the PM$_{2.5}$ concentration at a fixed location within the road section of Xinyao North Road at different times throughout the year using TSI 8530. We used ENVI-met to simulate the PM$_{2.5}$ concentration of the road section of Xinyao North Road (Model 1) during the same periods. Table 5 below shows the measured and simulated PM$_{2.5}$ concentrations. Error analysis was performed according to the measured values and simulated values with the coefficient of determination ($R^2$) and root mean square error (RMSE), in which $R^2$ was 0.779 ($p < 0.05$), and RMSE was 5.30 $\mu$g/m$^3$. Furthermore, through more actual measurement and simulation analysis, we found that ENVI-met can not only reasonably simulate the PM$_{2.5}$ concentration of the street canyon in different short periods of time (1–3 h) throughout the year but also has reasonable simulation accuracy for long-term simulation when the interference of other factors are suitably avoided. For example, the suburban trunk road with heavy traffic was selected for the study from around 19:00 in the evening to around 6 a.m. the next day to reduce the impact of other factors, such as PM$_{2.5}$ pollution from domestic and industrial sources on the street canyon. After conducting long-term continuous PM$_{2.5}$ simulation of traffic source (usually 8–12 h), the linear regression $R^2$ between the simulated values of ENVI-met and the actual reference values of PM$_{2.5}$ concentration in the street canyon is greater than 0.7 ($p < 0.05$), and the RMSE is usually less than 6 $\mu$g/m$^3$. It should be noted that the 0.3 m height was selected for the error analysis as it is close to the height of the car exhaust pipe [53,54]. PM$_{2.5}$ from traffic sources generally diffuses outward from motor vehicle exhaust pipes [55], and analysis at this height facilitates the analysis of error between the measured and simulated values at PM$_{2.5}$ generation height. In this study, except for the error analysis, the PM$_{2.5}$ concentration at the height of 1.5 m, which is close to the height of the human respiratory belt, has been used uniformly in the analysis of PM$_{2.5}$ concentrations in other parts of the study [56,57].

**Table 5.** Simulation values of the concentration of PM$_{2.5}$ at the measured point in each time period.

| Time | Measured PM$_{2.5}$ Concentration ($\mu$g/m$^3$) | Simulated PM$_{2.5}$ Concentration ($\mu$g/m$^3$) |
|---|---|---|
| 15:59:59 on 24 August 2022 | 26 | 27 |
| 17:59:59 on 24 August 2022 | 26 | 28 |
| 21:59:59 on 24 August 2022 | 34 | 36 |
| 9:59:59 on 28 September 2022 | 54 | 55 |
| 11:59:59 on 28 September 2022 | 49 | 50 |
| 15:59:59 on 28 September 2022 | 35 | 36 |
| 17:59:59 on 28 September 2022 | 36 | 36 |
| 16:59:59 on 22 December 2022 | 42 | 56 |
| 21:59:59 on 22 December 2022 | 34 | 42 |
| 16:59:59 on 24 December 2022 | 44 | 41 |

*3.4. Analysis of Simulation Results*

Figures 4 and 5 show the simulation results of the PM$_{2.5}$ concentration for Models 1–5 at 15:59:59 and 21:59:59 on 24th August 2022. The high-value areas (purple and pink areas in the black box) of PM$_{2.5}$ concentration greater than 30.60 $\mu$g/m$^3$ in Figure 4 and the high-value areas of PM$_{2.5}$ concentration greater than 37.50 $\mu$g/m$^3$ in Figure 5 all appear in the carriageway. The concentration of PM$_{2.5}$ decreases from the roadway to the sidewalk. This is obviously due to the fact that PM$_{2.5}$ in the study area is generated by motor vehicles on the carriageway and spreads out. The sidewalks on both sides of the carriageway are closer to the carriageway and have relatively high PM$_{2.5}$ concentrations.

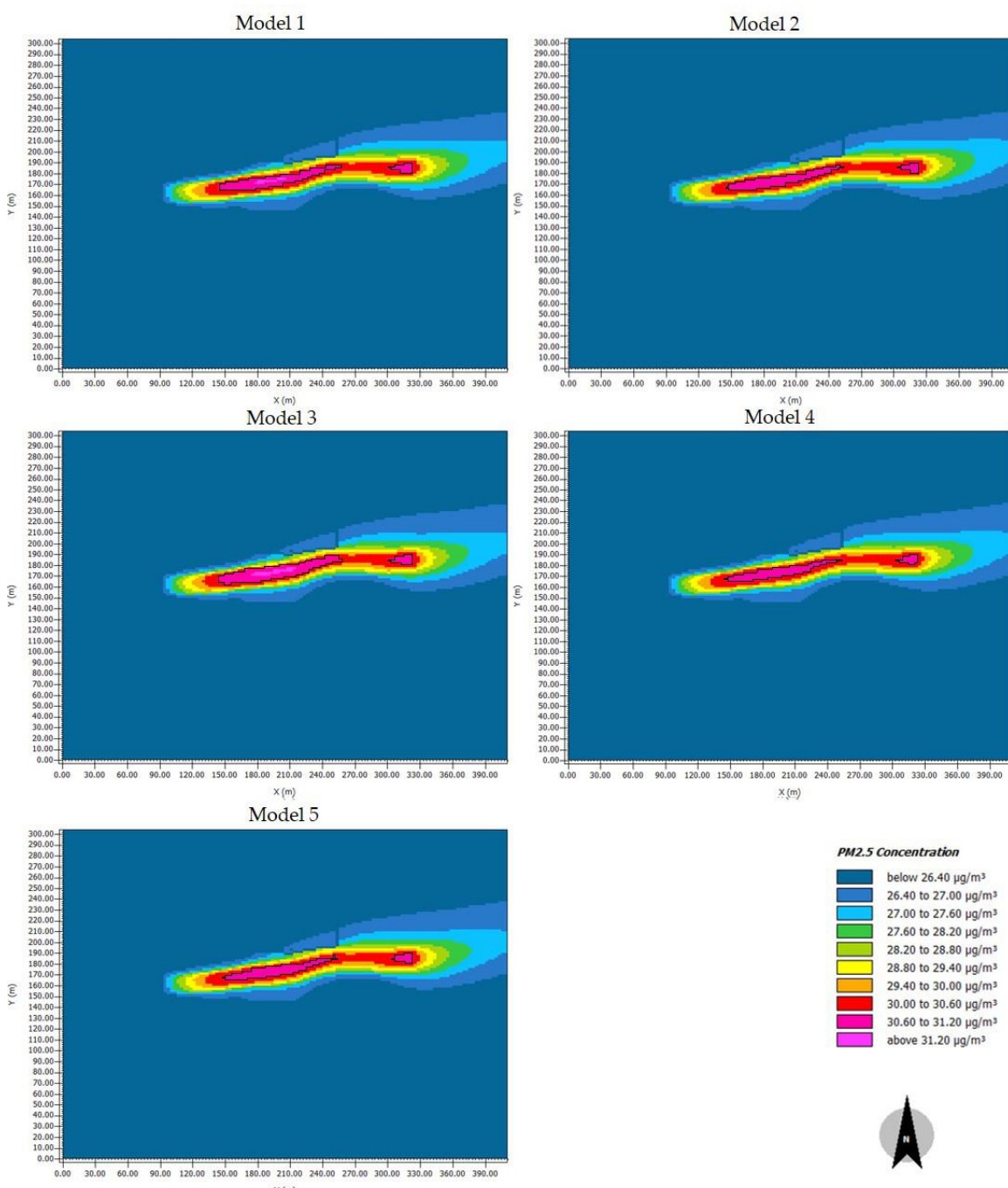

**Figure 4.** Simulation results of the concentration of PM$_{2.5}$ in five models at 15:59:59 on 24 August 2022.

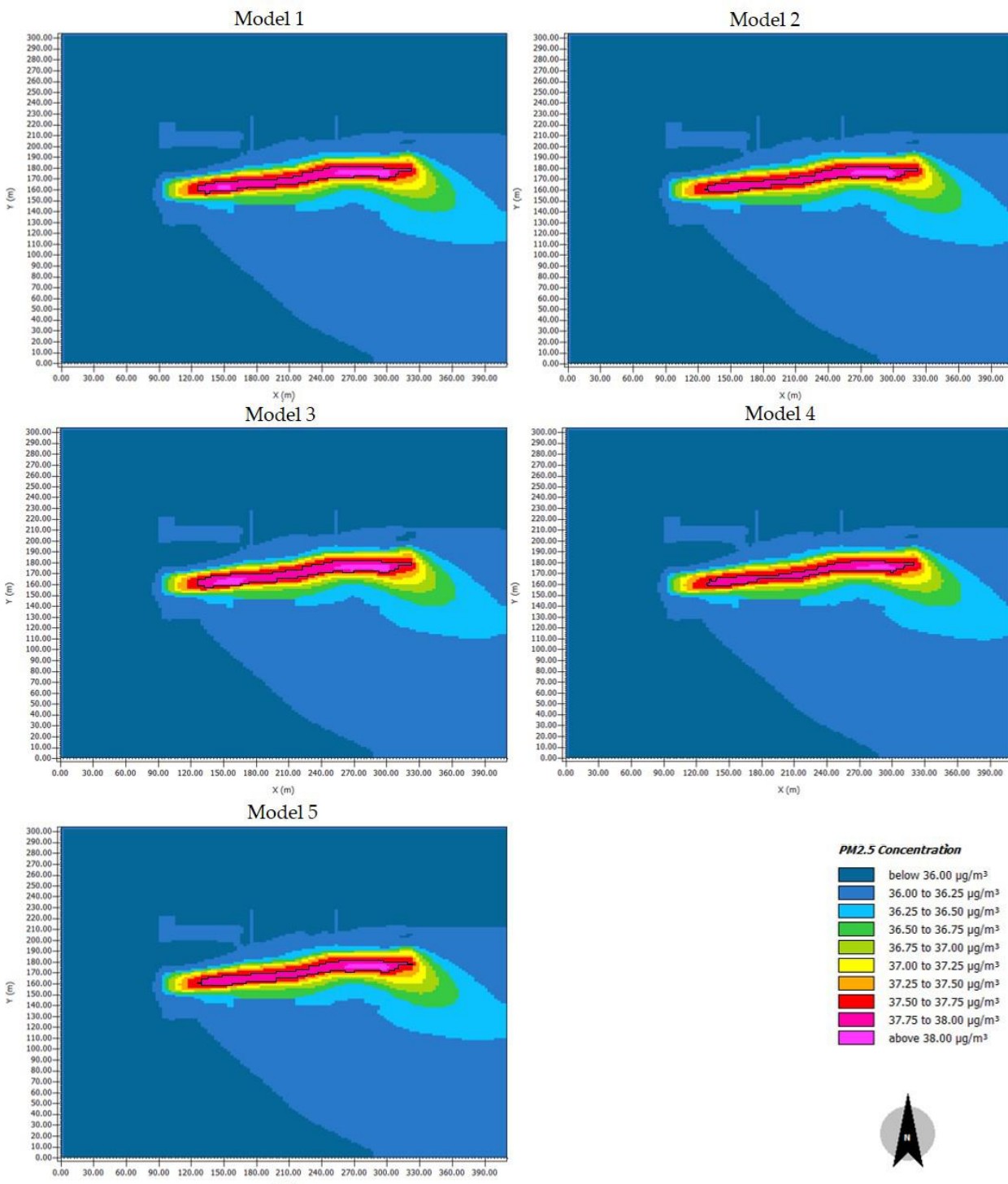

**Figure 5.** Simulation results of the concentration of PM$_{2.5}$ in five models at 21:59:59 on 24 August 2022.

We compared the areas indicated by the black-lined boxes of various models and found that the area of Model 3 is the largest and that of Model 4 is the smallest in the two time periods. Model 3 is constructed by adding more 5 m trees on the basis of the existing ten-meter street trees. This causes the roadside trees in this section to be closely arranged. Increasing the number of street trees is not conducive to ventilation, and the spread of PM$_{2.5}$ pollution and a large area of high PM$_{2.5}$ concentration appears in the region. There are many 5 m high street trees in Model 4. The canopy at 2–5 m can block the wind and

hinder the spread of $PM_{2.5}$. $PM_{2.5}$ pollution above 5 m can easily spread to the sidewalk and beyond without being blocked by street trees. Compared with the other four models in which the height of the street trees is mainly 10 m or 15 m, the area of the high concentration area of $PM_{2.5}$ on the carriageway in Model 4 is relatively small.

We selected 18 fixed coordinate points (six observation points on the roadway, six observation points on the sidewalk to the north of the roadway, and six observation points to the south of the roadway) for comparison. We extracted the $PM_{2.5}$ concentration of each coordinate point in each model. The average concentration values of the 18 observation points and the variance in the data are shown in Table 6. The number of street trees in Model 2 is less than that in Model 1. Model 3 is constructed by adding a certain number of 5 m high street trees on the basis of Model 1. By comparing the average concentrations of $PM_{2.5}$ at the corresponding points on the roadway in Model 1–3 (Table 6), it can be seen that reducing the number of street trees leads to a decrease in $PM_{2.5}$ concentrations in the area while increasing the number of street trees can increase the concentration of $PM_{2.5}$. It can be seen from Table 6 that the variance of the $PM_{2.5}$ concentration in Model 2 is 2.86. The variance of the $PM_{2.5}$ concentrations at each point in Model 1 is 2.92, which has the largest value and the largest dispersion degree, and the variance between the $PM_{2.5}$ concentrations at each point is the largest. The variance of the $PM_{2.5}$ concentrations in Model 3 is 2.91, which is slightly smaller than the variance corresponding to Model 1. The dispersion degree of this model ranks second. Further, adding street trees with a height of 5 m would lead to an overall increase in $PM_{2.5}$ concentrations in the study area, and the overall variance in $PM_{2.5}$ concentrations would be relatively large. If a certain number of street trees are removed, the overall $PM_{2.5}$ concentration in the area will be reduced in general, and the variance in the concentration of $PM_{2.5}$ will become smaller.

**Table 6.** The average and variance in $PM_{2.5}$ concentration values at 18 observation points in each model at 15:59:59 on 24 August 2022.

|  | Model 1 | Model 2 | Model 3 | Model 4 | Model 5 |
|---|---|---|---|---|---|
| Average ($\mu g/m^3$) | 28.09 | 28.06 | 28.10 | 28.05 | 28.06 |
| Variance | 2.92 | 2.86 | 2.91 | 2.80 | 2.85 |

When we compare the average $PM_{2.5}$ concentrations of the 18 observation points in the five models at 15:59:59 on 24 August 2022, we can see that the average $PM_{2.5}$ concentrations in Model 4 are lower than the values of the other four models (Table 6). The number of street trees in Model 4 is high among the five models, and their height is shorter than that in the other models. More 5 m high street trees may have a more positive impact on $PM_{2.5}$ concentrations in urban street canyons than fewer tall street trees. The number and location of street trees in Model 5 and Model 2 are the same, but the heights of the street trees differ. There is no significant difference between the influence of 10 m high street trees and 15 m high street trees on the surrounding $PM_{2.5}$ concentrations, and the average $PM_{2.5}$ concentration of the two models is 36.84 $\mu g/m^3$.

In order to determine whether there are different patterns during the day and at night, we simulated the relevant traffic flow and microclimate conditions based on the above Models 1–5 from 21:00 p.m. to 22:00 p.m. on 24th August 2022. We can see from Table 7 that the average and variance in $PM_{2.5}$ concentrations at all observation points in Model 2 are the lowest among Models 1–3, and the average of the $PM_{2.5}$ concentrations in Model 3 and Model 1 is equal. The variance in $PM_{2.5}$ concentration in Model 3 is larger than that in Model 1. The simulated $PM_{2.5}$ concentration in each model at 21:59:59 has the same rule as that of each model at 15:59:59; that is, the $PM_{2.5}$ concentration is generally lower in the model with fewer street trees, and the variance between $PM_{2.5}$ concentrations in the area is smaller on the premise that other conditions are consistent. The $PM_{2.5}$ concentrations in the model with more street trees are generally higher, and the differences between $PM_{2.5}$ concentrations in the region are larger. The average and variance in the $PM_{2.5}$ concentration

in Model 4 are the smallest, which indicates that the $PM_{2.5}$ concentration in the model with 5 m high street trees is the lowest, and the variance in the $PM_{2.5}$ concentration in the area is the smallest. The average and variance in the $PM_{2.5}$ concentrations for all observation points in Model 2 and Model 5 are equal, with an average value of 36.84 $\mu g/m^3$ and a variance of 0.42. This indicates that 10 m trees and 15 m trees have a similar influence on the surrounding environment. The simulation results show that there are no significant differences that emerged during the day and at night.

**Table 7.** The average and variance in $PM_{2.5}$ concentration at 18 observation points in each model at 21:59:59 on 24 August 2022.

|  | Model 1 | Model 2 | Model 3 | Model 4 | Model 5 |
|---|---|---|---|---|---|
| Average ($\mu g/m^3$) | 36.85 | 36.84 | 36.85 | 36.82 | 36.84 |
| Variance | 0.44 | 0.42 | 0.45 | 0.40 | 0.42 |

Figures 6 and 7 show the concentration of $PM_{2.5}$ at 11:59:59 on 28th September 2022 and at 15:59:59 on 28th September 2022, respectively. Figure 6 shows that the concentration of $PM_{2.5}$ is slightly higher in an area that is in the northwest of the models than in other off-road areas; the concentration of $PM_{2.5}$ in the northwest was mainly between 49.20 and 49.40 $\mu g/m^3$, while the concentration of $PM_{2.5}$ in the other off-road areas was below 49.20 $\mu g/m^3$. In Figure 7, the simulation results of each model show that the concentration of $PM_{2.5}$ in the southwest off-road area was between 35.30 and 35.60 $\mu g/m^3$, while the concentration of $PM_{2.5}$ in the other off-road areas was mainly below 35.30 $\mu g/m^3$. It is believed that this was caused by the wind. From 11:00 to 12:00, the wind direction was 122.6°. The southeasterly wind passed through the roadway and continued to blow further to the northwest. Therefore, there was an area with a high concentration of $PM_{2.5}$ in the northwest. From 15:00 to 16:00, the wind direction was 81.7°. The northeasterly wind passed through the roadway and continued to blow further to the southwest. Therefore, there was an area with a high concentration of $PM_{2.5}$ in the southwest. This supported the negative correlation between the mass concentration of $PM_{2.5}$ and wind direction suggested by Luo et al. (2021) [58].

By comparing the concentration of $PM_{2.5}$ in the roadway area in Figures 6 and 7, we can see that the purple and pink areas with a high concentration of $PM_{2.5}$ in Models 1 and 3 are relatively large, while the areas in Models 2, 4, and 5 are relatively small. In Model 2, the number of street trees was reduced on the basis of Model 1. Although Model 4 had more street trees than Model 1, the street trees in Model 4 were 5 m tall. Model 5 had fewer street trees than Model 1, but the trees in Model 5 were 15 m tall, which is taller than the 10 m street trees in Model 1. In Model 3, more 5 m street trees were added on the basis of Model 1. This indicates that reducing the number of street trees, rather than increasing the number, can reduce the concentration of $PM_{2.5}$ in an area with a high concentration of $PM_{2.5}$ on the roadway, supporting the conclusion of Guo et al. (2018) [59] that, to some extent, street trees cause more $PM_{2.5}$ to accumulate on the roadway, which is not conducive for its outward diffusion.

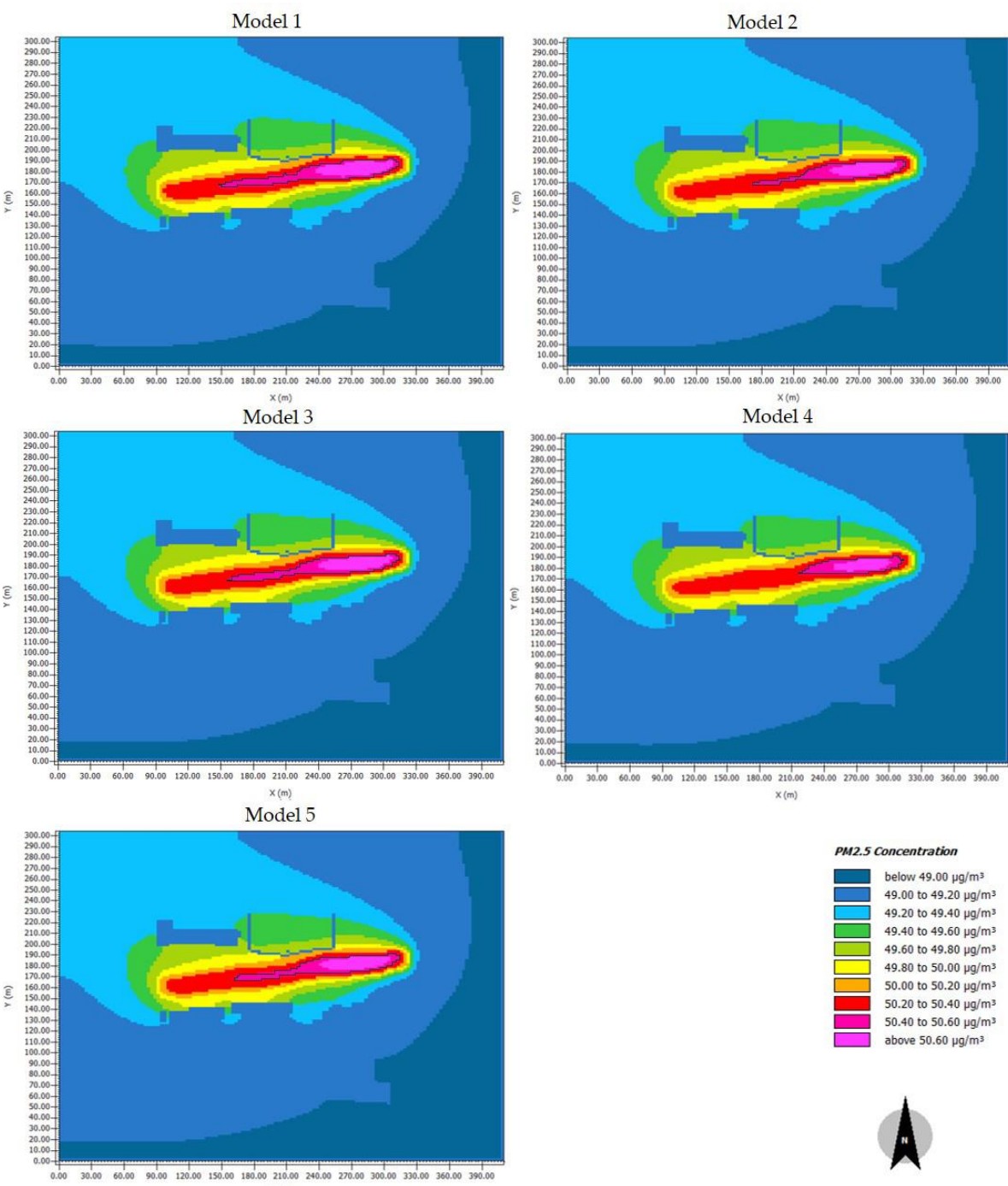

**Figure 6.** Simulation results of the concentration of PM$_{2.5}$ in five models at 11:59:59 on 28 September 2022.

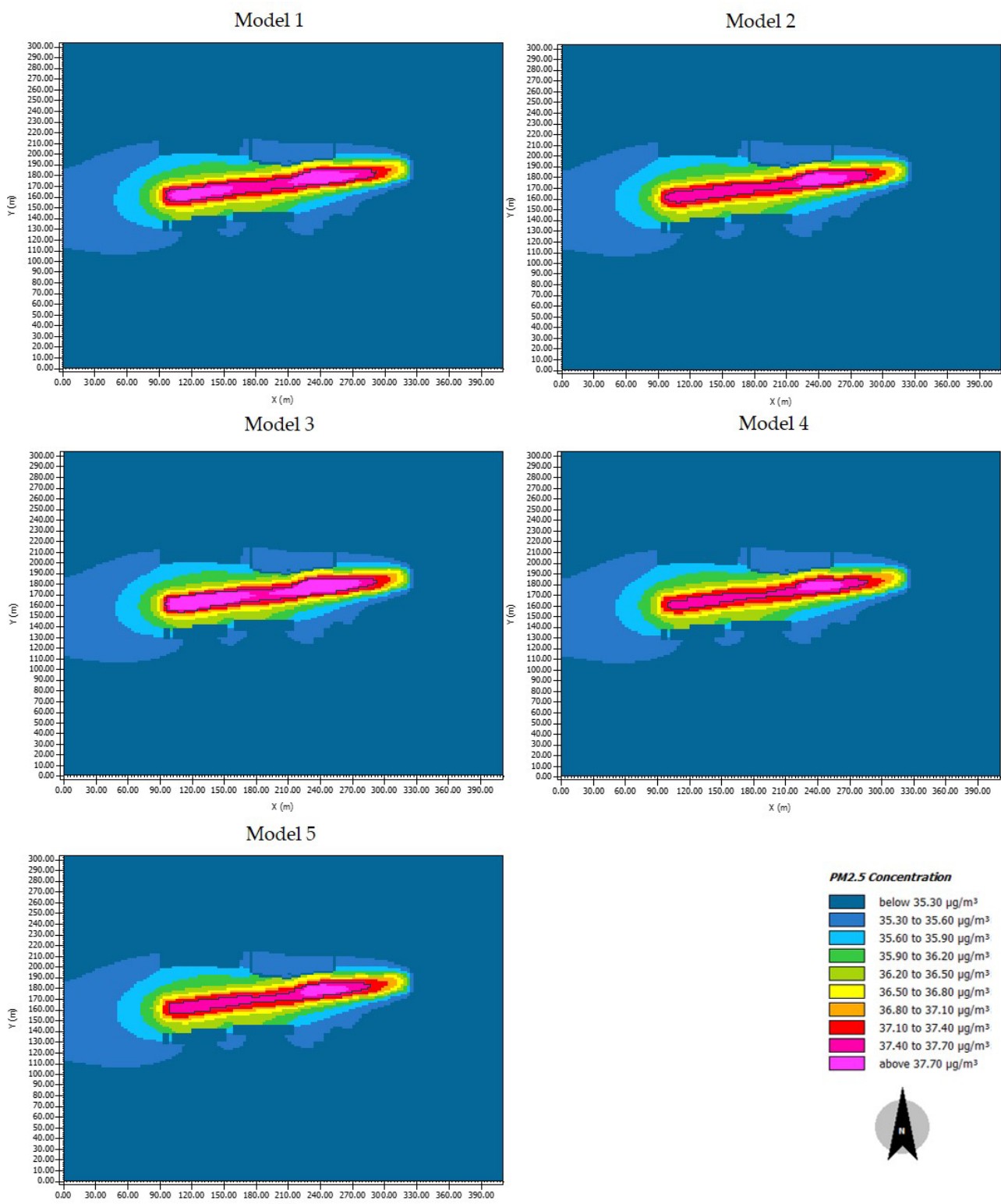

**Figure 7.** Simulation results of the concentration of PM$_{2.5}$ in five models at 15:59:59 on 28 September 2022.

The concentration of PM$_{2.5}$ at the same 18 coordinate points as in a previous study was extracted for comparison. Tables 8 and 9 list the average and variance in PM$_{2.5}$ concentrations at 11:59:59 and 15:59:59 on 28 September 2022, respectively, for the 18 observation points of the five models.

**Table 8.** The average and variance in PM$_{2.5}$ concentration values at 18 observation points in each model at 11:59:59 on 28 September 2022.

| | Model 1 | Model 2 | Model 3 | Model 4 | Model 5 |
|---|---|---|---|---|---|
| Average (µg/m$^3$) | 50.05 | 50.03 | 50.04 | 50.01 | 50.04 |
| Variance | 0.15 | 0.14 | 0.15 | 0.13 | 0.14 |

**Table 9.** The average and variance in PM$_{2.5}$ concentration values at 18 observation points in each model at 15:59:59 on 28 September 2022.

| | Model 1 | Model 2 | Model 3 | Model 4 | Model 5 |
|---|---|---|---|---|---|
| Average (µg/m$^3$) | 36.06 | 36.05 | 36.06 | 36.05 | 36.05 |
| Variance | 0.60 | 0.54 | 0.65 | 0.51 | 0.52 |

The concentration of PM$_{2.5}$ was negatively correlated with wind speed [16], indicating that the concentration of PM$_{2.5}$ in the street canyon would decrease faster with a higher wind speed. Different wind speeds would cause different street tree effects on the surrounding PM$_{2.5}$ concentration. The wind speeds from 11:00 to 12:00 and from 15:00 to 16:00 were 1 m/s and 1.26 m/s, respectively, which were relatively weak. A higher wind speed may amplify the difference in the concentrations of PM$_{2.5}$ between different models.

The average values of Model 4 in Tables 8 and 9 are 50.01 and 36.05, respectively, with variances of 0.13 and 0.51, respectively. These values are all less than or equal to the corresponding values in the other models. This model is characterized by low trees (5 m) and the smallest crown width (3 m). Even if there were more trees in the model than in many other models, the concentration of PM$_{2.5}$ on the roadway and sidewalk is the lowest. This also indicates that we should consider selecting short street trees with a small crown width when optimizing the pattern of street trees and giving full play to their role in reducing PM$_{2.5}$.

The average values of Model 2 in Tables 8 and 9 are 50.03 and 36.05, respectively, with variances of 0.15 and 0.60, respectively. The average concentration value is equal to or lower than that of Models 1, 3, and 5. The conditions of each street tree in Model 2 were the same as those in Model 1. The number of street trees in Model 2 was significantly lower than that in Model 1. This indicates that reducing the number of street trees is still beneficial for reducing the PM$_{2.5}$ concentration in street canyons, and more short trees may play a more positive role in the surrounding PM$_{2.5}$ concentration than fewer taller trees.

In Table 9, the average concentration of PM$_{2.5}$ in Model 3 was equal to that in Model 1, indicating that we may not be able to reduce the concentration of PM$_{2.5}$ on the roadway and sidewalk by adding more 5 m small trees between 10 m large trees [60]. By comparing the average concentration of PM$_{2.5}$ at 11:59:59 in Model 1 and Model 3, we found that the average concentration in Model 1 was higher than that in Model 3, indicating that continuing to increase the number of street trees on the basis of Model 1 has a slightly positive effect on the reduction in PM$_{2.5}$ pollution. This supports the conclusion of Buccolieri et al., (2018) [16] that street trees may have a positive effect on the surrounding PM$_{2.5}$ concentration due to the specific condition and the observation points or area people chose.

We also tried to carry out these simulations under different seasonal conditions to explore whether there are significant differences. We further conducted simulations based on the traffic flow and microclimate information of this road section in winter from 16:00 to 17:00 on 22 December 2022 and from 16:00 to 17:00 on 24 December 2022. Figures 8 and 9 show the simulated PM$_{2.5}$ concentration of each model at 16:59:59 on 22 December 2022 and at 16:59:59 on 24 December 2022, respectively. Like the previous four models, these two models were significantly affected by wind direction. The wind blew to the roadway from the southwest side of the model (wind direction 223.5°) and the northeast side (wind direction 45°) in the two periods. Figures 8 and 9 show areas with a relatively high PM$_{2.5}$ concentration on the northeast and southwest sides of each model. This also shows that

the wind was able to promote the diffusion of PM$_{2.5}$ pollution. We noticed this law in simulations of day and night, weekdays and weekends, and different seasons.

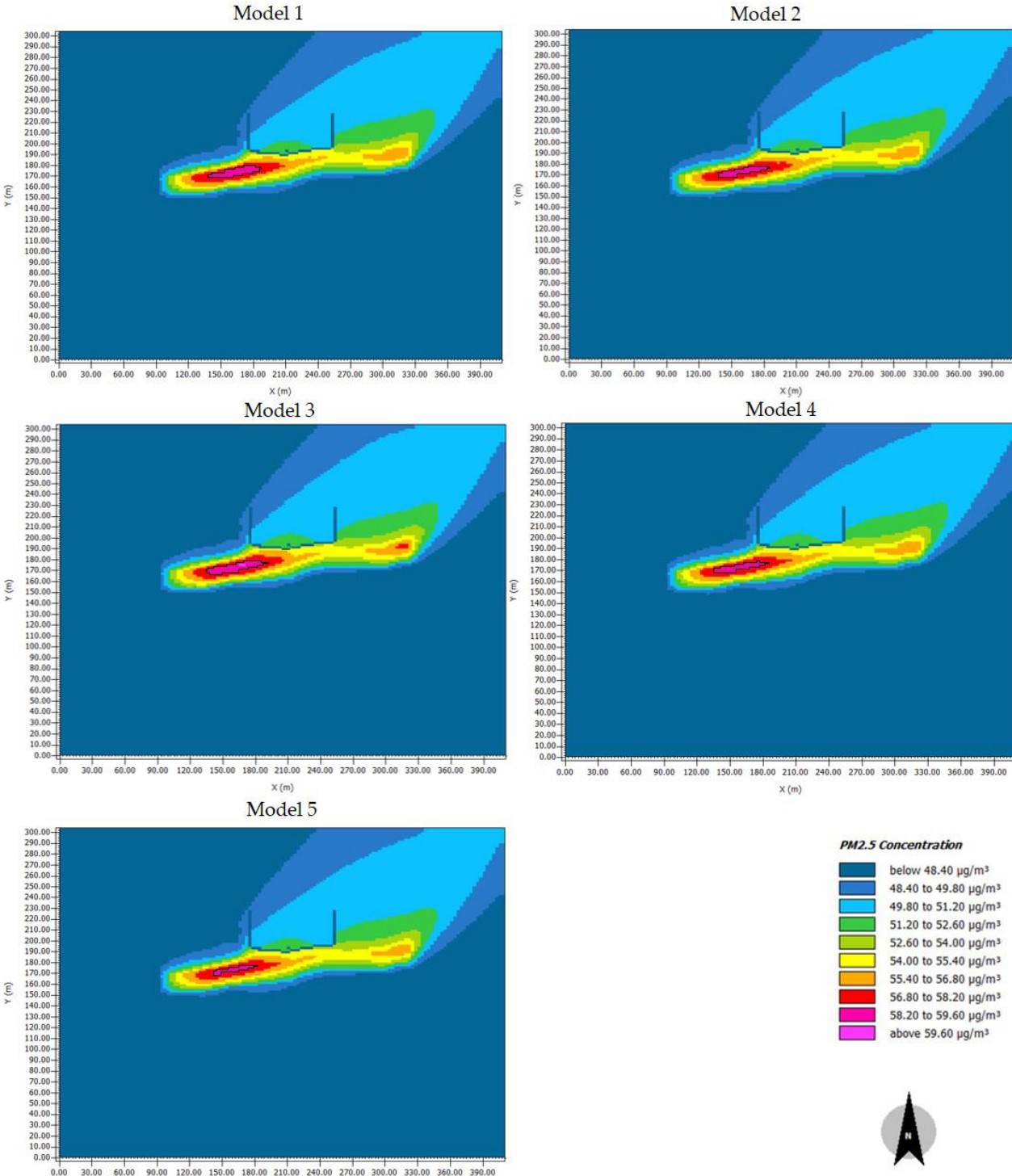

**Figure 8.** Simulation results of the concentration of PM$_{2.5}$ in five models at 16:59:59 on 22 December 2022.

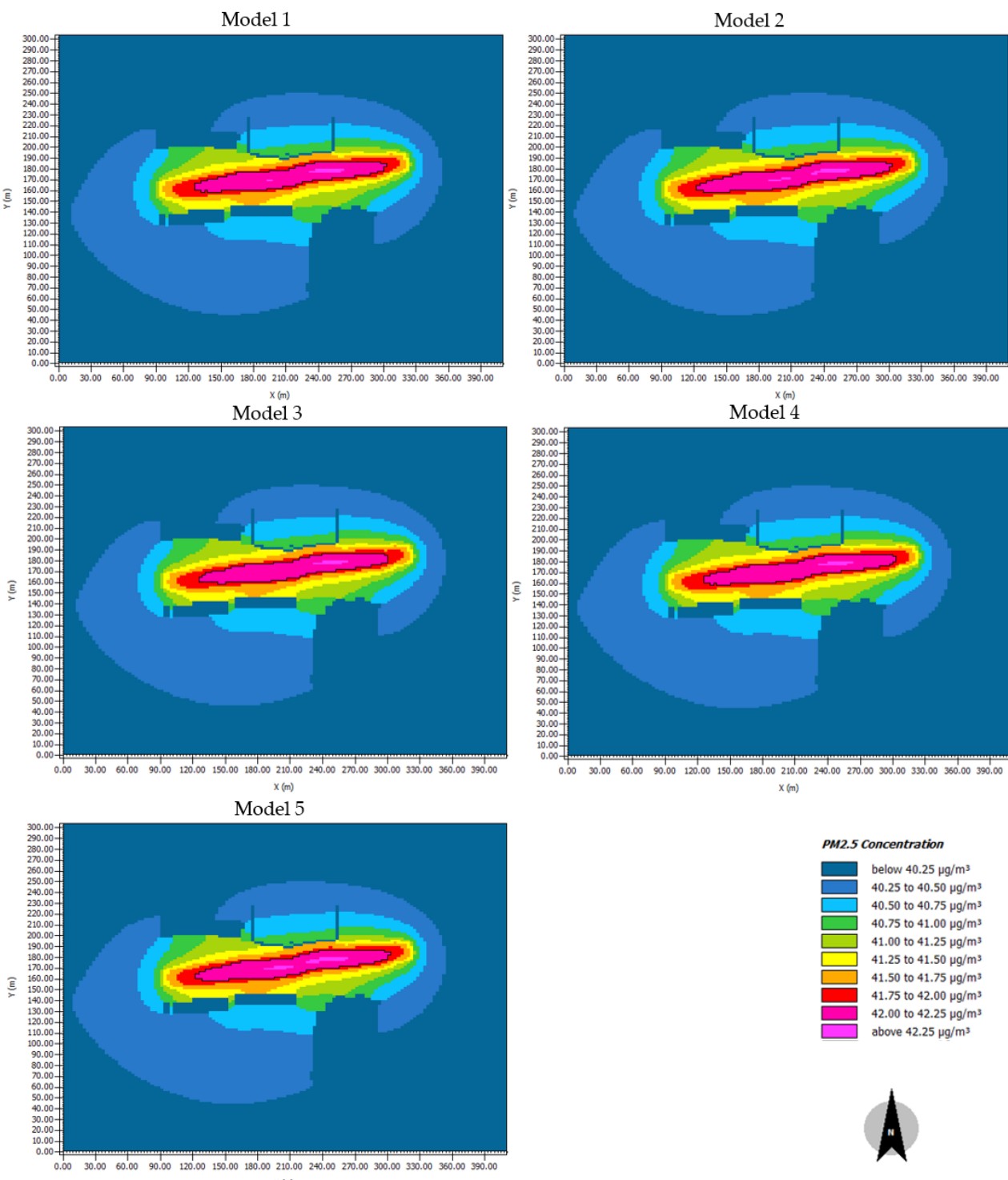

**Figure 9.** Simulation results of the concentration of PM$_{2.5}$ in five models at 16:59:59 on 24 December 2022.

We found that in the simulation results of the five models at 16:59:59 on December 22, the purple and pink areas of high PM$_{2.5}$ concentrations in Model 5, in which the street trees were mostly arbors with a height of 15 m, had the smallest area. This also showed that in this model, the area of high PM$_{2.5}$ pollution in the study area was relatively small. The areas of the purple and pink zones in Models 1–3 were as follows: Model 2 < Model 1 < Model 3. This meant that in the model where more 5 m street trees were added between 10 m street

trees on the basis of actual conditions, the area of the very high $PM_{2.5}$ concentration zone became larger. After reducing a certain amount of existing 10 m street trees, the area of the very high $PM_{2.5}$ concentration on the roadway in the model was reduced. The area of very high $PM_{2.5}$ pollution in Model 4 ranked third among the five models, which was lower than the corresponding areas in Model 3 and Model 1. The $PM_{2.5}$ concentration in the model with 5 m high street trees in the study area was relatively low in general.

In the simulation results of the five models at 16:59:59 on December 24, the area size of the $PM_{2.5}$ concentration high-value zone (purple and pink) in each model is in the order as follows: Model 5 > Model 2 > Model 1 > Model 4 > model 3. The number of street trees is greater, and the area of the $PM_{2.5}$ concentration extremely high-value zone in Model 4 with the tall street trees with a height of 5 m is no longer the lowest among the corresponding areas of the five models. Increasing the number of street trees based on the actual situation no longer results in a larger area of the $PM_{2.5}$ concentration high-value zone. Model 2 was formed by reducing the number of street trees based on the actual situation. The area of the $PM_{2.5}$ concentration high-value zone somewhat increases.

According to the simulation results for six different periods in three different seasons, we found that the gap between the highest and lowest values of $PM_{2.5}$ concentration in the study area was generally larger in the model with a higher wind speed. Table 10 displays the results of the simulations conducted at 16:00–17:00 on 22 December 2022; the average $PM_{2.5}$ concentrations of the 18 observation points in Model 2 were lower than those in Model 1. The average $PM_{2.5}$ concentrations in Model 3 were higher than those in Model 2. This was consistent with the law found for the previous models. Adding street trees based on the actual situation of the road section would lead to a higher overall $PM_{2.5}$ concentration in the study area. If the number of street trees was reduced, the $PM_{2.5}$ concentration in most areas of the models would be relatively low. During this period, the average $PM_{2.5}$ concentration of Model 4 was generally low. The mean and variance in $PM_{2.5}$ concentration of the 18 observation points of Model 4 were 52.10 and 9.42, respectively. The mean and variance in $PM_{2.5}$ concentration of Model 4 were lower than the corresponding values in Models 1–3. This shows that the $PM_{2.5}$ concentration was relatively low in the model with a street tree height of 5 m. The mean and variance in $PM_{2.5}$ concentration at each observation point in Model 5 (the mean was 50.04 and the variance was 9.09) were lower than those in the other four models. The $PM_{2.5}$ concentration at each observation point in the 15 m camphor tree model with a relatively small number was relatively low, and the degree of dispersion was small.

**Table 10.** The average and variance in $PM_{2.5}$ concentration values at 18 observation points in each model at 16:59:59 on 22 December 2022.

|  | **Model 1** | **Model 2** | **Model 3** | **Model 4** | **Model 5** |
|---|---|---|---|---|---|
| Average ($\mu g/m^3$) | 52.15 | 52.11 | 52.18 | 52.10 | 52.04 |
| Variance | 9.76 | 9.51 | 10.05 | 9.42 | 9.09 |

Table 11 shows the results of the simulations conducted at 16:00–17:00 on 24 December 2022; the overall difference in the average $PM_{2.5}$ concentration of the 18 observation points was small. The highest value (41.60) of the mean $PM_{2.5}$ concentration for each observation point of the five models was only 0.02 higher than the lowest one (41.58). The highest value (0.17) of the variance in $PM_{2.5}$ concentration for each observation point of the five models was only 0.01 higher than the lowest one. From the simulation results of the various models in the study area from 16:00 to 17:00 on 24 December 2022, it can be seen that Model 4 had relatively good simulation results on the whole (the mean was 41.59, and the variance was 0.16). This also shows that densely distributed with a relatively large number of 5 m high street trees was generally better. Our findings can guide subsequent relevant researchers to consider the relatively dense distribution of 5 m high street trees as a good practice that may produce better $PM_{2.5}$ concentrations in street canyons. The $PM_{2.5}$ concentration of Model

5 was higher than the corresponding values in the other four models. This also shows that some findings of relevant researchers may not apply to all street canyons because of different street canyon characteristics, plant species, and microclimate conditions, but the reference value of relevant studies is beyond doubt.

**Table 11.** The average and variance in PM$_{2.5}$ concentration values at 18 observation points in each model at 16:59:59 on 24 December 2022.

|  | **Model 1** | **Model 2** | **Model 3** | **Model 4** | **Model 5** |
|---|---|---|---|---|---|
| Average (μg/m$^3$) | 41.59 | 41.59 | 41.58 | 41.59 | 41.60 |
| Variance | 0.17 | 0.17 | 0.17 | 0.16 | 0.17 |

## 4. Discussion

In this study, we tried to explore how to optimize the street tree configuration so that PM$_{2.5}$ pollution in the street canyon can be minimized. Researchers built five different configuration models of street trees for the comparative experiment research. In this research, different seasons, days and nights, workdays and weekends, and other periods were considered to avoid the negative impact of focusing on a fixed period. By comparing the simulation results of the five models in six different periods throughout the year (each simulation lasted for one hour), we could identify that it was not true that the more street trees there were or the bigger the percentage of green space within the scope of the street canyon, the lower the concentration of PM$_{2.5}$ in the street canyon. Zhang et al. (2022) [61] studied the impact of green space landscape patterns on PM$_{2.5}$ within the urban area of Wuhan in Central China and found that the ratio of green space was inversely proportional to the concentration of PM$_{2.5}$. The research that is related to the scope of the street canyon and that of the completely different results for the entire city also indicate that green spaces might have different impacts on PM$_{2.5}$ in the environment at different scales. The research results and findings related to the scope of the street canyon cannot be directly used for other environments.

Whatever the selected time period and street tree pattern, PM$_{2.5}$ pollution always appears to have more obvious diffusion in the direction of the wind. Building geometry, tree morphology, and layout all significantly affect wind conditions and PM$_{2.5}$ dispersion. Under different weather conditions and vehicle numbers, the PM$_{2.5}$ concentration in the roadway area is always higher than that for the sidewalks on both sides.

By comprehensively considering the simulation results of six different periods, it can be seen that the average concentration of all the observation points in Model 2 is always equal to or lower than the corresponding average concentration in Model 1. The characteristics of a single street tree in Model 2 and Model 1 are exactly the same, and the number of street trees in Model 2 is much smaller than that in Model 1. There is a big gap between the street trees because there is a reasonable interval that can help with ventilation and improve air quality in the street canyon [62]. The dense canopies of trees can hinder ventilation and lead to the gathering of PM$_{2.5}$ in the environment of the street canyon [18].

In the five models, when there are many short and small 5 m street trees on both sides of the road, the distribution of the concentration of PM$_{2.5}$ in the street canyon is relatively better. This is a form of green space layout that should be considered first. This shows that the canopy and height of the street trees themselves should not be ignored when the impact of the interval between street trees on the concentration of PM$_{2.5}$ in the street canyon is considered. The crowns of 5 m street trees are significantly smaller than those of the 10 m and 15 m street trees. The change in the concentration of PM$_{2.5}$ in the surrounding area caused by street trees with a big canopy is bigger than that caused by street trees with a small canopy [18]. Street trees with a big canopy can have a more significant impact on air ventilation, leading to lower flow velocities and the gathering of pollutants around the street trees [63].

After comparing the simulation results of Model 2 and Model 3 at each period, it can be seen that in street canyons where 10 m high street trees are already densely arranged, the adding of many 5 m high street trees has a relatively weak negative impact on $PM_{2.5}$ concentration in the street canyon. Low plants have a relatively little negative impact on street canyons, which can be attributed to the factor that smaller crowns of 5 m high street trees have a relatively little impact on the dispersion of wind and $PM_{2.5}$.

In Model 4, where the height of the street trees is 5 m and there are many trees, the areas with high $PM_{2.5}$ values in the roadway area are relatively small. It is undoubtedly also very important to reduce the area of high $PM_{2.5}$ concentration in the roadway area. First of all, people sometimes open windows to ventilate the inside of their motor vehicle when driving. At this time, the $PM_{2.5}$ concentration on the roadway will have a relatively obvious influence on them. Additionally, if pedestrians are exposed to the area with the largest $PM_{2.5}$ in the street canyon when crossing the road, this area can also have a relatively serious negative influence on them. At this time, it is very important that the street trees in the area where the zebra crossing is located should not be too dense.

The location of the street tree trunks, tree species, and the number of trees in Model 2 and Model 5 are identical. The height of the street tree in Model 2 is 10 m, while this value is 15 m in Model 5. The main differences between the two models are reflected in the tree height, the starting height of the canopy, and the canopy (LAI, crown width, and crown height). The crown width of a 15 m high street tree is greater than that of a 10 m street tree. The distance between the crowns of two adjacent street trees in Model 2 is greater than that in Model 5. The starting height of the 15 m roadside tree canopy is greater than the value for 10 m street trees. The pollution level of $PM_{2.5}$ in Model 2 and Model 5 is similar, and the concentration of $PM_{2.5}$ in Model 2 is not usually lower than that in Model 5. Firstly, this indicates that, in this case, a larger distance between street trees does not mean it is easier for $PM_{2.5}$ dispersion This is because other factors can have an influence, like tree height, tree canopy width, and tree canopy height. In addition, the vertical distance between the canopy of a 15 m high street tree and the ground is relatively far, and its effect on the respiratory zone height of $PM_{2.5}$ will be weakened due to its distance from this height.

## 5. Conclusions

In this study, different street tree configuration models were constructed to explore how different factors such as the number, height, crown breadth, and spacing of street trees would have different influences on $PM_{2.5}$ concentrations in a street canyon. Moreover, this study explored whether the influence of street trees on $PM_{2.5}$ in a street canyon was significantly different under different meteorological conditions (wind direction, wind speed, temperature, humidity, sunlight, etc.) at different times in the year. Firstly, the dense distribution of too many street trees in the street canyon will lead to poor ventilation and a high $PM_{2.5}$ concentration in the street canyon. Therefore, compared with the continuously distributed street tree layout, the discontinuously distributed street tree layout is more helpful in reducing the $PM_{2.5}$ concentration in a street canyon through the dispersion of $PM_{2.5}$. Street trees with different heights and crown breadths have different influences on $PM_{2.5}$ pollution in a street canyon. Dwarf trees with a smaller crown breadth have less influence on $PM_{2.5}$ dispersion. Maintaining a reasonable spacing between street trees is conducive to $PM_{2.5}$ dispersion and optimizes the $PM_{2.5}$ concentration in a street canyon. The above findings did not differ significantly between days and nights, seasons, working days, and rest days. Based on the above factors, it can be seen from the high-value $PM_{2.5}$ areas on the roadway and from the average $PM_{2.5}$ concentration data of a large number of observation points that models with a street tree height of 5 m and that had a larger number of trees performed best overall. Street trees affect $PM_{2.5}$ in street canyons through complex settlement, stagnation, adsorption, and inhalation processes [64]. This study could not compare different models to determine the best spacing between street trees with the lowest $PM_{2.5}$ concentration in street canyons. However, we found through our research

that, for street trees with different heights and crown breadths, the spacing between them that can achieve better $PM_{2.5}$ dispersion will be different.

In this study, ENVI-met was used to analyze the distribution of $PM_{2.5}$ concentration in a street canyon under different street configurations. We cannot directly known from the simulation results how the street trees influence$PM_{2.5}$ concentration through sedimentation, retardation, adsorption, and inhalation. Further studies can be carried out to explore the differences in the sedimentation, retardation, adsorption, and inhalation generated from the presence of street trees with different numbers, heights, spacings, and canopy widths.

**Author Contributions:** Conceptualization, J.L. and B.Z.; methodology, J.L.; software, J.L.; validation, J.L. and B.Z.; formal analysis, J.L.; investigation, J.L. and B.Z.; resources, J.L.; data curation, J.L.; writing—original draft preparation, J.L.; writing—review and editing, J.L. and B.Z.; visualization, J.L.; supervision, B.Z.; project administration, J.L. and B.Z.; funding acquisition, J.L. and B.Z. All authors have read and agreed to the published version of the manuscript.

**Funding:** This research was funded by the Hunan Provincial Innovation Foundation for Postgraduate (grant number CX20230091), and Innovation Project for Postgraduates' Independent Exploration of Central South University (grant number: 2023ZZTS0001).

**Institutional Review Board Statement:** Not applicable.

**Informed Consent Statement:** Not applicable.

**Data Availability Statement:** Not applicable.

**Conflicts of Interest:** The authors declare no conflict of interest.

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
