# Peer review of "A Simulation Study on the Influence of Street Tree Configuration on Fine Particulate Matter (PM2.5) Concentration in Street Canyons"

_forests, doi:10.3390/f14081550_

Round 1

Reviewer 1 Report

Authors simulated dispersion of PM2.5 in street canyons ocuppied with trees. They applied ENVI-met model for that and conclude that lower treee (around 5m) aloow better air circulation and concentration of PM2.5 is lower when compared to tall trees. Authors should make the papert substantially shorter. You show too many graphs and tables. Paper should go directyl to the point and do not show everything you created. In addtion, the sentces should be shorter, currently the text is vague (writing almost a page that Kestrel sensor is OK to use has no sense).  I am missing discussion too! Discussion is very important to compare what you have found with what others found and where is the novelty of this paper. So where is the novelty? What you are showing here is already known. Please add discussion and point out the novelty of the paper.

Here are specific comments:

page 1: it is not 2,5 microns, but 2.5 μm. Make it correct

"Only when the concentration of PM2.5 is equal to 0 will it not be harmful to the human body." - write better English, I think this statement is obvious, do not write it here

Page 2: "In general, there are few studies on the effects of street greening on the surrounding PM2.5 concentration" see paper 10.1016/j.ufug.2022.127757 - it shows the effect of greenery on urban air quality for PM10 and how that is remover. It has similar implications even for PM2.5

what is it H/W,LADs please define

2.2 add citation to ENVI-met

page 12: define LAI, you define later that you use it

table 6: having a table with only one line is very strange. Rather add this to the text

Discussion: missing

Conclusion: should be shorter

Too long and vague sentences with rather simple English.

Author Response

Suggestion 1: Authors simulated dispersion of PM2.5 in street canyons ocuppied with trees. They applied ENVI-met model for that and conclude that lower treee (around 5m) aloow better air circulation and concentration of PM2.5 is lower when compared to tall trees. Authors should make the papert substantially shorter. You show too many graphs and tables. Paper should go directyl to the point and do not show everything you created. In addtion, the sentces should be shorter, currently the text is vague (writing almost a page that Kestrel sensor is OK to use has no sense).  I am missing discussion too! Discussion is very important to compare what you have found with what others found and where is the novelty of this paper. So where is the novelty? What you are showing here is already known. Please add discussion and point out the novelty of the paper.

Comment 1: We have made the paper substantially shorter. We have added a discussion part. Please see the “3.5 discussions” in the article. We discussed the novelty of the paper in the discussion part. In addition, we have rewritten the abstract to show the novelty of the paper in the abstract.

Here are specific comments:

Suggestion 2: page 1: it is not 2,5 microns, but 2.5 μm. Make it correct

Comment 2: Thanks for your suggestion. We have changed from 2,5 microns to 2.5 μm.

Suggestion 3: "Only when the concentration of PM2.5 is equal to 0 will it not be harmful to the human body." - write better English, I think this statement is obvious, do not write it here

Comment 3: We have rewritten the sentence as " However small the concentration of PM2.5 is in the air, it can still negatively affect the human body. ". We have deleted the original sentence.

Suggestion 4: Page 2: "In general, there are few studies on the effects of street greening on the surrounding PM2.5 concentration" see paper 10.1016/j.ufug.2022.127757 - it shows the effect of greenery on urban air quality for PM10 and how that is remover. It has similar implications even for PM2.5
Comment 4: We are sorry. We were thinking to stress that the overall number of research on the influence of road green space on PM2.5 in street canyons is not very large, and more research needs to be conducted to better learn the influence of green space landscape patterns on PM2.5 in street canyons. We have rewritten this sentence as "Some researchers have conducted research on the influence of road green space on particulate matter. This article attempts to further explore the number, height, spacing, and canopy characteristics (Crown Height and Crown Breadth, etc.) of street trees on PM2.5 concentrations in street valleys based on existing research."

Suggestion 5: what is it H/W, LADs please define

Comment 5: We have defined H/W as aspect ratio and LADs as leaf area density.

Suggestion 6: 2.2 add citation to ENVI-met

Comment 6: We have added citations corresponding to the introduction to ENVI-met in Section 2.2

Suggestion 7: page 12: define LAI, you define later that you use it

Comment 7: We are sorry. We have specified the abbreviation of leaf area index as LAI when it appears for the first time in the article, and we use LAI in the subsequent part to replace its full name. 

Suggestion 8:  table 6: having a table with only one line is very strange. Rather add this to the text

Comment 8: We have deleted Table 6 and then added the corresponding content into the text.

Suggestion 9:  Discussion: missing

Comment 9: We have added a discussion part (3.5 part in the article).

Suggestion 10:  Conclusion: should be shorter

Comment 10: We have rewritten the conclusion.

Suggestion 11: Comments on the Quality of English Language

Too long and vague sentences with rather simple English.

Comment 11: We have revised the whole article for the language.

Reviewer 2 Report

The manuscript "A Simulation Study on the Influence of Street Tree Configuration on PM2.5 Concentration in Street Canyons" study PM2.5 in different heights and distances of street trees. The ENVI-met model was used to simulate results which is very interesting.

However, there are some suggestions to improve the quality of the work.

1) Extensive editing of English language is required, especially the writing style. In many parts, the writing can be shorten with the same meaning.

2) PM2.5 should be written as "PM subscript 2.5"

3) Many sentences need revision since they cannot be understood. For example, "We found that the wind is able to promote the diffusion of PM2.5 in the street canyon to the wind direction" "In most case, increasing the number of street trees is not conductive to ventilation ..."

4) There are many typos in the manuscript. For example, weuse in stead of We used; 2.7 Error Analysis should be in another line; reference 46 in the reference list has no information.

5) What is "dust" in "Statistical data from 2018 revealed that dust ..."? Is it resuspended dust?

6) There is no motorcycle in the study area? If there are motorcycles, why they are not included in the study?

7) What are the differences between model 1, 2, 3, 4 and 5 (figure 4,5, ...8)? It may be better to summarize in Table and highlighted in Figures. The texts can be removed -- too much.

8) Is it possible to link Figure 9&10 with Figure 4-8? Maybe using the same scale or direction. It is difficult to understand.

9) Table 13: Line for 15:59:59 seems to be in the wrong place

10) Why the instrument to measure PM2.5 was placed at 0.3m? Usually, it should be close to the breathing level.

11) The conclusion should be shortened and more precise.  

Extensive editing of English language is required, especially the writing style. In many parts, the writing can be shorten with the same meaning.

Author Response

The manuscript "A Simulation Study on the Influence of Street Tree Configuration on PM2.5 Concentration in Street Canyons" study PM2.5 in different heights and distances of street trees. The ENVI-met model was used to simulate results which is very interesting.

However, there are some suggestions to improve the quality of the work.

Suggestion 1: 1) Extensive editing of English language is required, especially the writing style. In many parts, the writing can be shorten with the same meaning.

Comment 1: We are sorry. We tried to shorten the sentences by ourselves at first. In addition, we used the English editing service. We hope the article is concise and clear now.

Suggestion 2:  2) PM2.5 should be written as "PM subscript 2.5"

Comment 2: Thanks quite a lot for your suggestion. We have changed all PM2.5 to PM2.5 (2.5 is the subscript form) in the whole article.

Suggestion 3:  3) Many sentences need revision since they cannot be understood. For example, "We found that the wind is able to promote the diffusion of PM2.5 in the street canyon to the wind direction" "In most case, increasing the number of street trees is not conductive to ventilation ..."

Comment 3: We are sorry. We have revised the whole article.

Suggestion 4:  4) There are many typos in the manuscript. For example, weuse in stead of We used; 2.7 Error Analysis should be in another line; reference 46 in the reference list has no information.

Comment 4: We have revised the typos.

Suggestion 5:  5) What is "dust" in "Statistical data from 2018 revealed that dust ..."? Is it resuspended dust?

Comment 5: Yeah, it is resuspended dust. We are sorry for the unclear expression. We have changed it to resuspended dust.

Suggestion 6:  6) There is no motorcycle in the study area? If there are motorcycles, why they are not included in the study?

Comment 6 : Motorcycle riding is prohibited in the road segment we study in Changsha, so there is no motorcycle in this area. Besides, to avoid the impact of electric bicycles (which do not belong to vehicles that should be simulated in ENVI-met), we selected road segments and periods with few electric bicycles.

Suggestion 7:  7) What are the differences between model 1, 2, 3, 4 and 5 (figure 4,5, ...8)? It may be better to summarize in Table and highlighted in Figures. The texts can be removed -- too much.

Comment 7: We have removed the texts, summarize the information in a table and highlighted in figures. You can see the explanation about the highlights in the paragraph before the figures.

Suggestion 8:  8) Is it possible to link Figure 9&10 with Figure 4-8? Maybe using the same scale or direction. It is difficult to understand.

Comment 8: We've combined Figures 4-8 into a big picture (Please note that the picture numbers have changed). Models 1-5 in Figure 3 (new number) are schematic diagrams of each model respectively. Models 1-5 in Figures 4-6 (new numbers) are simulated results of each model in each time period. Please note that Figures 4-6 was generated by the simulation software, which uniformly generated the picture with north at the top. In Fig. 3, we want to better show the situation of street trees in street canyons. We have cropped a 3D sketch up picture. So this picture does not feature north at the top. We hope this will make the picture convey relevant information more clearly.

Suggestion 9:  9) Table 13: Line for 15:59:59 seems to be in the wrong place

Comment 9: We are so sorry. We have deleted this misplaced line and redrawn a line at the correct position.

Suggestion 10:  10) Why the instrument to measure PM2.5 was placed at 0.3m? Usually, it should be close to the breathing level.

Comment 8: The height of car exhaust pipe is generally around 0.3m, and the PM2.5 from traffic source is usually generated by car exhaust. The purpose of the actual measurement is to verify the error by comparing it with the corresponding simulated value. This height is conducive to generating a highly visual comparison of the error between the measured values and the simulated values from PM2.5. As PM2.5 concentration values at different heights can be simulated in ENVI-met, we only use the measured values at 0.3m with the simulated values in the error analysis. In this study, except for the error analysis, during the analysis of other PM2.5 concentrations, we have uniformly selected the data at the height of 1.5m of the breathing zone for analysis. We have explained this issue in Section 3.3 Error Analysis.

Suggestion 11:  11) The conclusion should be shortened and more precise.  

Comment 11: We have rewritten the conclusion to make it shorter and more precise.

Comments on the Quality of English Language

Suggestion 12:  Extensive editing of English language is required, especially the writing style. In many parts, the writing can be shorten with the same meaning.

Comment 12 : We have extensively edited the English language.

Round 2

Reviewer 1 Report

I thank authors for the changes they did.

Still some vague sentences with odd wordings are here. Otherwise the English is better now.

Author Response

Comments and Suggestions for Authors

I thank authors for the changes they did.

Thanks for participating in the review process and for your valuable suggestions.

Comments on the Quality of English Language

Suggestion 1: Still some vague sentences with odd wordings are here. Otherwise the English is better now.

Comment 1: We have repeatedly reread the manuscript and tried to improve language expression.

Reviewer 2 Report

Thank you for your revision. It is shorten, but the manuscript is till too long. Some simple English should not be used in the publication, such as figure out in "In order to figure out how...". I still would like to request the authors to make the paper more concise. Too many words which can be shorten. 

Here are some suggestions:

1. PM2.5 is PM subscript 2.5 (old comments and still in the current version)

2. English needs extensive revision. Most sentences and Table can be shorten. For example, Table 1: Why the authors need to repeat the word "Road section of ....adjadent..." in every line?

3. What is a "and a reasonable range or measurement error" on page 6? Also, some similar unclear information are still in the manuscript. 

4. Why these road sections were selected for simulation? Please explain what are reasons and what are differences?

Extensive editing of English language required.

Author Response

Suggestion 1: Thank you for your revision. It is shorten, but the manuscript is till too long. Some simple English should not be used in the publication, such as figure out in "In order to figure out how...". I still would like to request the authors to make the paper more concise. Too many words which can be shorten. 

Comment 1: Thanks for participating in the review process and for your valuable suggestions. We have repeatedly reread the manuscript and tried to improve language expression.

Here are some suggestions:

Suggestion 2: 1. PM2.5 is PM subscript 2.5 (old comments and still in the current version)

Comment 2: We have checked the full text and changed all 2.5 of PM2.5 to the subscript format. By the way, we found that after adjusting all PM2.5 to PM subscript 2.5, when we open the document again, some paragraph fonts will automatically change and then the subscripts in some paragraphs will disappear.

Suggestion 3: 2. English needs extensive revision. Most sentences and Table can be shorten. For example, Table 1: Why the authors need to repeat the word "Road section of ....adjadent..." in every line?

Comment 3: We are sorry. We have tried to shorten the article with more precise language.

Suggestion 4: 3. What is a "and a reasonable range or measurement error" on page 6? Also, some similar unclear information are still in the manuscript. 

Comment 4: We have added information for the sentences that provide unclear information.

Suggestion 5: 4. Why these road sections were selected for simulation? Please explain what are reasons and what are differences?

Comment 5: We have explained the reasons and showed the difference in 2.3. Field survey part and 3.2. Building the models part. Especially the second paragraph of the 2.3 part and the first and second paragraph of the 3.2 part.